# A Better Baseline for Second Order Gradient Estimation in Stochastic Computation Graphs

## Abstract

Motivated by the need for higher order gradients in multi-agent reinforcement learning and meta-learning, this paper studies the construction of baselines for second order Monte Carlo gradient estimators in order to reduce the sample variance. Following the construction of a *stochastic computation graph* (SCG), the *Infinitely Differentiable Monte-Carlo Estimator* (DiCE) can generate correct estimates of arbitrary order gradients through differentiation. However, a baseline term that serves as a control variate for reducing variance is currently provided only for first order gradient estimation, limiting the utility of higher-order gradient estimates. To improve the sample efficiency of DiCE, we propose a new baseline term for higher order gradient estimation. This term may be easily included in the objective, and produces unbiased variance-reduced estimators under (automatic) differentiation, without affecting the estimate of the objective itself or of the first order gradient. Importantly it reuses the same baseline function (e.g. the state-value function in reinforcement learning) already used for the first order baseline. We provide theoretical analysis and numerical evaluations of our baseline term, which demonstrate that it can dramatically reduce the variance of second order gradient estimators produced by DiCE. This computational tool can be easily used to estimate second order gradients with unprecedented efficiency wherever automatic differentiation is utilised, and has the potential to unlock applications of higher order gradients in reinforcement learning and meta-learning.

## 1 Introduction

Problems that have intractable stochasticity often give rise to objectives that are not directly differentiable. In reinforcement learning, for example, the expected return involves an expectation over the stochasticity induced by both the policy and the environment dynamics. As a result, the gradient of this objective with respect to the policy parameters cannot be directly calculated. However, we may construct Monte Carlo estimates of the gradient from samples, and then apply gradient-based optimisation methods. In such problems, the score function trick (Fu, 2006) may be used to easily define estimates of the first order gradients of the stochastic objective. However, to calculate these gradient estimates in practice, we need to leverage the powerful toolbox of automatic differentiation. To this end, Schulman et al. (2015) introduce the *surrogate loss* (SL) within the formalism of *stochastic computation graphs* (SCG). Under single differentiation the SL produces an unbiased gradient estimator for the first order gradient of the primary objective.

Estimating higher order gradients of these stochastic objectives is important in many settings. One example is in multi-agent learning. In *Learning with Opponent-Learning Awareness* (LOLA) (Foerster et al., 2018a), the expected return of one agent is differentiated through the learning step of another agent. Another use-case is the combination of meta-learning with gradient-based methods applied to reinforcement learning (Finn et al., 2017; Al-Shedivat et al., 2017), as the objective function is repeatedly differentiated through a hierarchy of learning processes. Lastly, higher order gradients may be used directly for some optimisation algorithms, such as the quasi-Newton algorithm (Wright & Nocedal, 1999).

Despite the success of higher order gradients in these applications, accurate and efficient estimation of higher order gradients is challenging. This is due to the complexity of constructing the corresponding estimators, as well as their extremely high variance. It is therefore critical to develop

methods that produce low variance estimates of higher order gradients and are easy to use in practice through automatic differentiation. One way to reduce variance of stochastic estimators is by introducing control variates (Paisley et al., 2012; Tucker et al., 2017; Grathwohl et al., 2017). The key contribution of this work is the design of a variance reduction technique for higher order gradient estimators based on the control variate framework and amenable to automatic differentiation.

Given an objective function with random variables, the naive approach is to derive the corresponding higher order gradient estimators analytically. Al-Shedivat et al. (2017) use the score function estimator repeatedly to obtain proper estimators for higher orders in a meta-learning setting. Foerster et al. (2018a) combine this approach with Taylor expansion to generate higher order gradient estimators in a multi-agent setting. However, this approach is difficult to generalise to arbitrary objective functions, as well as being error-prone and incompatible with automatic differentiation.

Schulman et al. (2015) argue that the gradient estimate produced by differentiating an SL objective may be treated as an objective itself, and a new SL can be built to estimate higher order gradients. Unfortunately, as shown by Foerster et al. (2018b), this approach can produce incorrect higher order gradient estimates because the cost terms are treated as fixed samples in the SL, and so it does not maintain all necessary dependencies in the SCG after differentiation. Foerster et al. (2018b) propose the *Infinitely **Di**fferentiable Monte-Carlo Estimator* (DiCE) to estimate higher order gradients correctly while maintaining the ease of use by leveraging automatic differentiation.

DiCE uses two methods to reduce the variance of its first order gradient estimates. First, it respects the causal dependence of costs on stochastic events that can influence them, and second, it implements a simple baseline method. However, the higher order gradient estimates generated by DiCE still have high variance, and their quality relies on a large number of samples.

In this paper, we design a novel baseline term for second-order gradient estimation based on DiCE. This term can be easily included in the original objective without changing the estimate of the objective itself or of the first order gradients. Nonetheless, appropriate control variates for the higher order gradients are produced automatically upon repeated differentiation of the modified objective. In this baseline term, we include an additional dependence which properly captures the causal relationship between stochastic nodes in an SCG. Using the DiCE approach, we are able to preserve these dependencies through differentiation. We prove that adding our baseline term to the DiCE objective will not affect the expected estimate of second order gradients. We explicitly derive the second order gradients of the DiCE objective with our baseline term in order to demonstrate how it introduces control variates for higher order gradient estimates. Additionally, we conduct a series of numerical evaluations to verify the correctness of our baseline term and show its effectiveness for variance reduction. We believe that our baseline term is an important missing component to make DiCE an extremely powerful and valuable tool to access higher order gradients in reinforcement learning and meta-learning.

## 2 BACKGROUND

Suppose $x$ is a random variable distributed as $x \sim p(x; \theta)$, and $f(x)$ is a deterministic function of $x$. We assume that $f(x)$ is independent of $\theta$ so that $\nabla_\theta^n f = 0$ for $n \in \{1, 2, \dots\}$. Suppose we have the objective function $\mathcal{L}(\theta) = \mathbb{E}_x[f(x)]$ and we need to compute the gradients of the expectation, $\nabla_\theta \mathbb{E}_x[f(x)]$, in order to use gradient-based optimization methods. An unbiased gradient estimator $g$ is a random variable such that $\mathbb{E}[g(f)] = \nabla_\theta \mathbb{E}[f(x)]$. The *score function estimator* (Fu, 2006) is an unbiased estimator given by

$$\mathbb{E}_x[f(x)\nabla_\theta \log p(x; \theta)] = \nabla_\theta \mathbb{E}_x[f(x)]. \tag{1}$$

The control variates method is a variance reduction technique for improving the efficiency of Monte Carlo estimators. A *control variate* is a function $c(x)$ whose value can be easily obtained, and whose expectation $\mathbb{E}[c(x)]$ is known. Using the control variate, we can construct a new random variable

$$z(f) = g(f) - c(x) + \mathbb{E}_x[c(x)]. \tag{2}$$

The expectation of the random variable $z(f)$ is

$$\mathbb{E}[z(f)] = \mathbb{E}_x[g(f)] - \mathbb{E}_x[c(x)] + \mathbb{E}_x[c(x)] = \mathbb{E}_x[f(x)].$$

Consequently, if $g(f)$ is an unbiased estimator of the gradient, so is $z(f)$. However, if the random variables $g(f)$ and $z(f)$ are positively correlated, the variance of $z(f)$ can be lower than that of $g(f)$ (Grathwohl et al., 2017).

## 2.1 STOCHASTIC COMPUTATION GRAPHS

A *stochastic computation graph* (SCG) (Schulman et al., 2015) is a directed and acyclic graph that consists of three types of nodes:

1. **Input nodes** $\Theta$**:** an input node $\theta \in \Theta$ contains parameters set externally. We may be interested in the dependence of an objective function on these nodes, and will attempt to differentiate the objective with respect to their parameters.

2. **Deterministic nodes** $\mathcal{D}$**:** a deterministic node $d \in \mathcal{D}$ is a deterministic function of its parent nodes.

3. **Stochastic nodes** $\mathcal{S}$**:** a stochastic node $w \in \mathcal{S}$ is a random variable whose distribution is conditioned on its parent nodes.

Additionally, cost nodes $\mathcal{C}$ can be added into the formalism of an SCG without loss of generality. A cost node $c \in \mathcal{C}$ is a determinstic function of its parent nodes that produces a scalar value. The set of cost nodes $\mathcal{C}$ are those associated with an objective function $\mathcal{L} = \mathbb{E}[\sum_{c \in \mathcal{C}} c]$. In an SCG, $(v, w)$ is a directed edge that connects node $v$ and non-input node $w$, where $v$ is a parent node of $w$. The notation $v \prec w$ means there is a path from node $v$ to node $w$, i.e., node $v$ influences node $w$.

## 2.2 DICE

In order to estimate higher order gradients correctly, Foerster et al. (2018b) propose the *Infinitely Differentiable Monte-Carlo Estimator* (DiCE) within the formalism of SCGs. DiCE uses the magic box $\boxdot$ as a novel operator. The input to $\boxdot$ is a set of stochastic nodes $\mathcal{W}$, and $\boxdot$ is designed to have the following properties:

1. $\boxdot(\mathcal{W}) \rightarrowtail 1$,
2. $\nabla_\theta \boxdot(\mathcal{W}) = \boxdot(\mathcal{W}) \sum_{w \in \mathcal{W}} \nabla_\theta \log p(w; \theta)$.

Here $\rightarrowtail$ means "evaluates to", which is different from "equals to", $=$, i.e., full equality including equality of all derivatives. In the context of a computation graph, $\rightarrowtail$ denotes a forward pass evaluation. In contrast, the second property describes the behaviour of $\boxdot$ under differentiation. The right hand side of this equality can in turn be evaluated to estimate gradients as described below, or differentiated further. To achieve these properties, Foerster et al. (2018b) show that $\boxdot$ can be straightforwardly implemented as follows:

$$\boxdot(\mathcal{W}) = \exp(\tau - \perp(\tau)),$$

where $\tau = \sum_{w \in \mathcal{W}} \nabla_\theta \log p(w; \theta)$ and $\perp$ is a 'stop-grad' operator that sets the derivative to zero, $\nabla_x \perp(x) = 0$, as it is commonly available in auto-differentiation libraries. This implementation will give us the required properties of the magic box operator $\boxdot$ in practice.

For a node $w$ in an SCG, we use $\mathcal{S}_w$ to denote the set of stochastic nodes that influence the node $w$ and are influenced by $\theta$, i.e., $\mathcal{S}_w = \{s | s \in \mathcal{S}, s \prec w, \theta \prec s\}$. Using $\boxdot$, the DiCE objective (Foerster et al., 2018b) is defined as

$$\mathcal{L}_{\boxdot} = \sum_{c \in \mathcal{C}} \boxdot(\mathcal{S}_c) c. \tag{3}$$

Under repeated differentiation the DiCE objective generates arbitrary order gradient estimators (Foerster et al., 2018b, Theorem 1): $\mathbb{E}[\nabla_\theta^n \mathcal{L}_{\boxdot}] \rightarrowtail \nabla_\theta^n \mathcal{L}$, for $n \in \{0, 1, 2, \dots\}$.

## 2.3 VARIANCE REDUCTION WITH DICE

$\mathcal{L}_{\boxdot}$ by itself already implements a simple form of variance reduction by respecting causality. In gradient estimates, each cost node $c$ is multiplied by the sum of gradients of log-probabilities of

only upstream nodes $\mathcal{S}_c$ that can influence $c$. This reduces variance compared to using the log joint probability of *all* stochastic nodes, which would still create an unbiased gradient estimate.

However, Foerster et al. (2018b) offer additional variance reduction for first-order gradient estimation by including a baseline

$$\mathcal{B}_{\boxdot}^{(1)} = \sum_{w \in \mathcal{S}} (1 - \boxdot(\{w\})) b_w, \tag{4}$$

where $b_w$ [1] is a function of the set NONINFLUENCED$(w) = \{v | w \nprec v\}$, i.e. the set of nodes that does not influence $w$. Here $b_w$ may be chosen to reduce variance, and a common choice for $b_w$ is the average cost-to-go, i.e. $\mathbb{E}[R_w | \text{NONINFLUENCED}(w))]$. Here $R_w = \sum_{c \in \mathcal{C}_w} c$, where $\mathcal{C}_w = \{c | c \in \mathcal{C}, w \prec c\}$, i.e., the set of cost nodes that depend on node $w$.[2] In order to maintain unbiased gradient estimates, the baseline factor $b_w$ should be a function that is independent of the stochastic node $w$. Greensmith et al. (2004) provide an overview of variance reduction techniques for gradient estimators, including the use of this type of baseline. The baseline term $\mathcal{B}_{\boxdot}^{(1)}$ can be added to the DiCE objective to obtain

$$\mathcal{L}_{\boxdot}^{b_1} = \mathcal{L}_{\boxdot} + \mathcal{B}_{\boxdot}^{(1)}. \tag{5}$$

Note that $\mathcal{L}_{\boxdot}^{b_1}$ evaluates to the same value as $\mathcal{L}_{\boxdot}$ because $(1 - \boxdot(\mathcal{W})) \rightarrowtail 0$.

## 3 METHOD

First, consider explicitly the effect of the traditional baseline on the first order gradient estimates. The estimates without and with baseline are as follows (derivations given in Appendix A.1):

$$\nabla_\theta \mathcal{L}_{\boxdot} \rightarrowtail \sum_{w \in \mathcal{S}} R_w \nabla_\theta \log p(w; \theta),$$

$$\nabla_\theta \mathcal{L}_{\boxdot}^{b_1} \rightarrowtail \sum_{w \in \mathcal{S}} (R_w - b_w) \nabla_\theta \log p(w; \theta). \tag{6}$$

For each stochastic node $w \in \mathcal{S}$, the term $b_w \nabla_\theta \log p(w; \theta)$ works as a control variate to reduce the variance of the term $R_w \nabla_\theta \log p(w; \theta)$. To ensure the appropriate correlations, $b_w$ may be a function trained to estimate $\mathbb{E}[R_w | \text{NONINFLUENCED}(w))]$. As a result, $\nabla_\theta \mathcal{L}_{\boxdot}^{b_1}$ is a first order gradient estimator with lower variance than $\nabla_\theta \mathcal{L}_{\boxdot}$. Additionally, $\nabla_\theta \mathcal{L}_{\boxdot}^{b_1}$ is unbiased because $\mathbb{E}[\nabla_\theta \mathcal{B}_{\boxdot}^{(1)}] \rightarrowtail 0$ (see Appendix A.1).

Note that in (6) we have omitted a term, $\sum_{c \in \mathcal{C}} \nabla_\theta c$, that arises when the cost nodes depend directly on $\theta$. In most use-cases this term will not appear, as the costs are sampled from an unparameterised process, such as the unkown environment in reinforcement learning. This straight-through contribution to the gradient estimate is also typically much lower variance than the contribution estimated using the score function trick. Due to these considerations, we assume that the costs are independent of $\theta$ in the remainder of this work, although the remaining terms for both first and second order are derived in the appendix.

Next, we consider second-order gradient estimation using the objective $\mathcal{L}_{\boxdot}^{b_1}$. The second order gradient of the DiCE objective can be evaluated as follows (derivations given in Appendix A.2):

$$\nabla_\theta^2 \mathcal{L}_{\boxdot} \rightarrowtail \sum_{w \in \mathcal{S}} R_w \frac{\nabla_\theta^2 p(w; \theta)}{p(w; \theta)} + 2 \sum_{w \in \mathcal{S}} \nabla_\theta \log p(w; \theta) \left[ \sum_{v \in \mathcal{S}, w \prec v} R_v \nabla_\theta \log p(v; \theta) \right],$$

$$\nabla_\theta^2 \mathcal{L}_{\boxdot}^{b_1} \rightarrowtail \sum_{w \in \mathcal{S}} (R_w - b_w) \frac{\nabla_\theta^2 p(w; \theta)}{p(w; \theta)} + 2 \sum_{w \in \mathcal{S}} \nabla_\theta \log p(w; \theta) \left[ \sum_{v \in \mathcal{S}, w \prec v} R_v \nabla_\theta \log p(v; \theta) \right]. \tag{7}$$

---

[1] b(NONINFLUENCED$(w)$) in (Schulman et al., 2015)

[2] We use the $R_w$ notation to correspond with a *return* for readers familiar with reinforcement learning, a key use case. The cost notation for nodes is kept to maintain consistency with Schulman et al. (2015).

The baseline objective function $\mathcal{L}_{\boxdot}^{b_1}$ still implements a partial variance reduction in the second order gradient estimates, by providing a control variate for the first term in (7). However, the $R_v$ in the second term are not paired with suitable control variates. As a result, the variance of this term could be extremely high. In fact, due to the nested summations over the high-variance $R_v$, this term can dominate the variance of total gradient estimate. We explore this empirically in section 4, where we observe that $\mathcal{B}_{\boxdot}^{(1)}$ is of little use for reducing the overall variance of the second order gradient estimates.

### 3.1 A Second Order Baseline

To substantially reduce variance for the second order gradient estimator, we propose a new baseline:

$$\mathcal{B}_{\boxdot}^{(2)} = \sum_{w \in \mathcal{S}'} \left(1 - \boxdot(\{w\})\right) \cdot \left(1 - \boxdot(\mathcal{S}_w)\right) b_w \tag{8}$$

Here $\mathcal{S}' = \{w | w \in \mathcal{S}, \mathcal{S}_w \neq \emptyset, \theta \prec w\}$, i.e., the set of stochastic nodes that depend on $\theta$ and at least one other stochastic node. Note that $\mathcal{B}_{\boxdot}^{(2)} \rightarrowtail 0$ because $(1 - \boxdot(\{w\})) \rightarrowtail 0$ and $(1 - \boxdot(\mathcal{S}_w)) \rightarrowtail 0$, and $b_w$ is the same as that used in $\mathcal{B}_{\boxdot}^{(1)}$. The new DiCE objective function becomes:

$$\mathcal{L}_{\boxdot}^{b_2} = \mathcal{L}_{\boxdot} + \mathcal{B}_{\boxdot}^{(1)} - \mathcal{B}_{\boxdot}^{(2)}. \tag{9}$$

Since $\mathcal{B}_{\boxdot}^{(1)} \rightarrowtail 0$ and $\mathcal{B}_{\boxdot}^{(2)} \rightarrowtail 0$, $\mathcal{L}_{\boxdot}$, $\mathcal{L}_{\boxdot}^{b_1}$, and $\mathcal{L}_{\boxdot}^{b_2}$ all evaluate to the same estimate of the original objective. Further, all derivatives of our modified objective $\mathcal{L}_{\boxdot}^{b_2}$ are unbiased estimators of the derivatives of the original objective, that now contain suitable control variates for variance reduction. We now show how this baseline term indeed leads to a lower variance while still being an unbiased estimate for the higher order derivatives of our objective.

### 3.2 Bias and Variance Analysis

In the DiCE objective $\mathcal{L}_{\boxdot}$, for each cost node $c \in \mathcal{C}$, the corresponding $\boxdot(\mathcal{S}_c)$ reflects the dependency of $c$ on all stochastic nodes which influence it (and depend on $\theta$). In contrast, the baseline term $\mathcal{B}_{\boxdot}^{(1)}$ only includes $\boxdot(\{w\})$, considering each stochastic node $w$ separately. This simple approach results in variance reduction for first order gradients, as shown in (6). However, the failure to capture the dependence of stochastic nodes on each other in the simple baseline prevents it from reducing the variance of the cross terms that arise in second order derivatives (the final term in (7)). To capture these relationships properly, we include $\boxdot(\mathcal{S}_w)$ in the definition of $\mathcal{B}_{\boxdot}^{(2)}$, i.e., an additional dependence on the stochastic nodes that influence $w$. Use of the $\boxdot$ operator ensures that these dependencies are preserved through differentiation.

To verify that our proposed baseline indeed captures these dependencies appropriately, we now consider its impact on the gradient estimates. The first and second order gradients of the baseline term $\mathcal{B}_{\boxdot}^{(2)}$ can be evaluated as follows (derivations given in Appendix A.2):

$$\nabla_\theta \mathcal{B}_{\boxdot}^{(2)} \rightarrowtail 0,$$
$$\nabla_\theta^2 \mathcal{B}_{\boxdot}^{(2)} \rightarrowtail 2 \sum_{w \in \mathcal{S}} \nabla_\theta \log p(w; \theta) \Big[ \sum_{v \in \mathcal{S}, w \prec v} b_v \nabla_\theta \log p(v; \theta) \Big]. \tag{10}$$

The first order gradient estimates remain unchanged: as $\nabla_\theta \mathcal{B}_{\boxdot}^{(2)} \rightarrowtail 0$, $\nabla_\theta \mathcal{L}_{\boxdot}^{b_2}$ evaluates to the same value as $\nabla_\theta \mathcal{L}_{\boxdot}^{b_1}$. The second order gradient estimate of our full objective, $\nabla_\theta^2 \mathcal{L}_{\boxdot}^{b_2}$, is as follows:

$$\nabla_\theta^2 \mathcal{L}_{\boxdot}^{b_2} \rightarrowtail \sum_{w \in \mathcal{S}} (R_w - b_w) \frac{\nabla_\theta^2 p(w; \theta)}{p(w; \theta)} + 2 \sum_{w \in \mathcal{S}} \nabla_\theta \log p(w; \theta) \Big[ \sum_{v \in \mathcal{S}, w \prec v} (R_v - b_v) \nabla_\theta \log p(v; \theta) \Big]. \tag{11}$$

Control variates have been introduced for the terms in the second part of (11), when using our new baseline term $\mathcal{B}_{\boxdot}^{(2)}$. $R_w$ and $b_w$ are positively correlated by design, as they should be for variance

reduction of the first order gradients. As a result, the estimator $\nabla_\theta^2 \mathcal{L}_{\square}^{b_2}$ could have significantly lower variance compared with $\nabla_\theta^2 \mathcal{L}_{\square}^{b_1}$ and $\nabla_\theta^2 \mathcal{L}_{\square}$, as we verify empirically in Section 4.

Furthermore, we verify that our baseline does not change the expected estimate of second order derivatives.

**Theorem 1.** $\mathbb{E}\left[\nabla_\theta^2 \mathcal{L}_{\square}^{b_2}\right] \rightarrowtail \nabla_\theta^2 \mathbb{E}\left[\mathcal{L}\right]$.

*Proof.* First, we can prove that $\mathbb{E}[\nabla_\theta^2 \mathcal{B}_{\square}^{(1)}] \rightarrowtail 0$ and $\mathbb{E}[\nabla_\theta^2 \mathcal{B}_{\square}^{(2)}] \rightarrowtail 0$ (See Appendix). Since $\nabla_\theta^2 \mathcal{L}_{\square}$ is an unbiased estimator of $\nabla_\theta^2 \mathbb{E}\left[\mathcal{L}\right]$, i.e., $\nabla_\theta^2 \mathcal{L}_{\square} \rightarrowtail \nabla_\theta^2 \mathbb{E}\left[\mathcal{L}\right]$, then:

$$\mathbb{E}\left[\nabla_\theta^2 \mathcal{L}_{\square}^{b_2}\right] = \mathbb{E}\left[\nabla_\theta^2 \mathcal{L}_{\square}\right] + \mathbb{E}\left[\nabla_\theta^2 \mathcal{B}_{\square}^{(1)}\right] - \mathbb{E}\left[\nabla_\theta^2 \mathcal{B}_{\square}^{(2)}\right] \rightarrowtail \nabla_\theta^2 \mathbb{E}\left[\mathcal{L}\right].$$

$\square$

Thus, $\nabla_\theta^2 \mathcal{L}_{\square}^b$ is an unbiased second order gradient estimator of the original objective $\mathbb{E}\left[\mathcal{L}\right]$.

## 3.3 REINFORCEMENT LEARNING

We now consider the particular case of reinforcement learning. Given a policy $\pi$, we can generate an episode of horizon $T$:

$$\tau = (s_0, a_0, r_0, \ldots, s_T, a_T, r_T).$$

The discounted return at time step $t$ is the discounted sum of future rewards, $R_t(\tau) = \sum_{k=t}^T \gamma^{k-t} r_t$, where $\gamma \in [0,1]$ is a discount factor. When the reinforcement learning problem is formalised as an SCG, the cost nodes are the discounted rewards and the objective function is $\mathcal{L} = \mathbb{E}[\sum_{t=0}^T \gamma^t r_t]$. The corresponding DiCE objective function is:

$$\mathcal{L}_{\square} = \sum_{t=0}^T \square(a_{t' \leq t}) \cdot \gamma^t r_t, \tag{12}$$

where $a_{t' \leq t}$ is the set of all previous actions at time step $t$, i.e., $a_{t' \leq t} = \{a_0, a_1, \ldots, a_t\}$. Clearly, these are the stochastic nodes that influence the reward at time $t$. We choose the baseline $b(s_t)$ to be a function of state $s_t$; it must be independent of the action $a_t$. In particular, we choose $b(s_t) = \gamma^t \hat{V}(s_t)$, where $\hat{V}(s_t)$ is an estimate of the state value function $V^\pi(s_t) = \mathbb{E}[R_t|s_t]$. First order variance reduction may now be achieved with the baseline term:

$$\mathcal{B}_{\square}^{(1)} = \sum_{t=0}^T (1 - \square(a_t)) b(s_t). \tag{13}$$

To reduce the variance of the second order gradient estimators, we can use our novel baseline term:

$$\mathcal{B}_{\square}^{(2)} = \sum_{t=1}^T \left(1 - \square(a_t)\right) \cdot \left(1 - \square(a_{t' < t})\right) b(s_t). \tag{14}$$

These baseline terms can be added to our original objective. As in the general case, the corresponding DiCE objectives with baselines are $\mathcal{L}_{\square}^{b_1} = \mathcal{L}_{\square} + \mathcal{B}_{\square}^{(1)}$ and $\mathcal{L}_{\square}^{b_2} = \mathcal{L}_{\square} + \mathcal{B}_{\square}^{(1)} - \mathcal{B}_{\square}^{(2)}$.

In $\square(a_{t' < t})$, we need to have strict inequality $t' < t$, which captures the causality from all previous actions. The agent is able to look backward at its past actions but excludes its current action. Since there is no previous action at $t = 0$, the summation runs from $t = 1$ to $t = T$. It is essential to exclude the current action $a_t$ at time step $t$ for variance reduction because the baseline $b(s_t)$ must be independent of the action $a_t$ to remain unbiased. Note that this does not leave a term in the gradient estimate without a control variate: the "diagonal" term corresponding to only the action at $t = 0$ is already addressed by the second derivative of $\mathcal{B}_{\square}^{(1)}$ and appears in the first term in (11).

## 4 EXPERIMENTS

First, we numerically verify that DiCE with our new baseline term $\mathcal{L}_{\square}^{b_2}$ can generate correct estimators of the Hessians in an SCG using a set of randomly initialised fixed policies in the *iterated prisoner's dilemma*.

In this setting, two agents play the game of the prisoner's dilemma iteratively. At each round, there are two possible actions for each agent, which are Cooperate (C) and Defect (D). As a result, there are four possible outcomes, CC, CD, DC, and DD at each round, which are the observation at the next time step. The payoff matrix of this game is given in Figure 1.

**Multi-agent DiCE.** The objective function for agent $i$ is $\mathcal{L}^i = \mathbb{E}[\sum_{t=0}^{T} \gamma^t r_t^i]$. The per-agent DiCE objective $\mathcal{L}_{\square}^i$ is a simple extension of (12), replacing $r_t$ by the per-agent reward $r_t^i$, and $a_t$ by the joint action $a_t^{j \in \{1,2\}}$. For correct higher order gradients it is essential to consider the dependence of the reward on the actions of both agents in this way. We will require per-agent baseline factors $b^i(s_t)$ to then form per-agent baseline terms $\mathcal{B}_{\square}^{i,(1)}$ and $\mathcal{B}_{\square}^{i,(2)}$ in analogy with (13, 14). Again, the single-agent action at each timestep is replaced by the joint action.

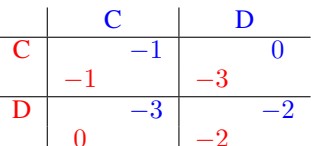

Figure 1: The payoff matrix of prisoner's dilemma. Numbers in a cell correspond to the utilities of the player with the same colour.

Using DiCE, the dependencies between the returns and parameters of the two agents are accounted for, and first and second order gradients can be estimated efficiently using automatic differentiation. We will test and compare the performance of $\mathcal{L}_{\square}^{b_1,i}$ and $\mathcal{L}_{\square}^{b_2,i}$ in second-order gradient estimation.

Foerster et al. (2018a) derive the value function of IPD analytically, which we use as ground truth to verify the correctness of our estimator. Figure 2(b) shows that we can obtain correct second-order gradient estimates using our baseline term. Comparing to Figure 2(a), our novel baseline term dramatically improves the estimation of second order gradients. In the original DiCE paper (Foerster et al., 2018b), a sample size of 100k is required to obtain second order gradient estimates with correlation coefficient 0.97. After formulating our baseline term, the required sample size is reduced to 1k, a reduction by two orders of magnitude. Figure 3 shows the correlation coefficients of the exact Hessian and the estimated Hessian using different sample sizes. These results demonstrate that our baseline term is important for estimating second order gradients accurately and efficiently when using DiCE.

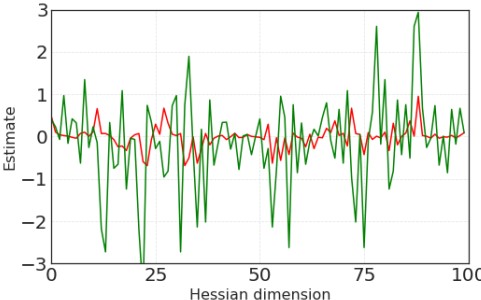
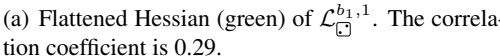
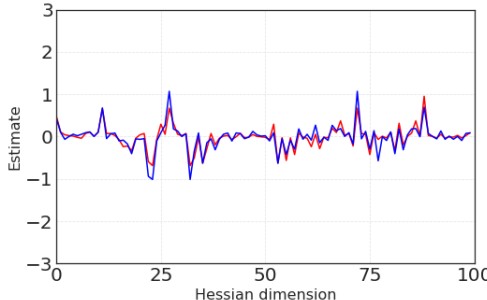

(a) Flattened Hessian (green) of $\mathcal{L}_{\square}^{b_1,1}$. The correlation coefficient is 0.29.

(b) Flattened Hessian (blue) of $\mathcal{L}_{\square}^{b_2,1}$. The correlation coefficient is 0.97.

Figure 2: Flattened true (Red) and estimated Hessian of agent 1 for the iterated prisoner's dilemma. The sample size is 1000.

**LOLA-DiCE.** In LOLA-DiCE (Foerster et al., 2018b) agents differentiate through the learning step of other agents, using the DiCE objective to calculate higher order derivatives:

$$\mathcal{L}^1(\theta_1, \theta_2)_{\text{LOLA}} = \mathbb{E}_{\pi_{\theta_1}, \pi_{\theta_2} + \Delta\theta_2(\theta_1, \theta_2)}[\sum_{t=0}^{T} \gamma^t r_t^1],$$

where $\Delta\theta_2(\theta_1, \theta_2) = \alpha_2 \nabla_{\theta_2} \mathbb{E}_{\pi_{\theta_1}, \pi_{\theta_2}}[\sum_{t=0}^{T} \gamma^t r_t^2]$ and $\alpha_2$ is a step size. Training details are provided in Appendix A.3.

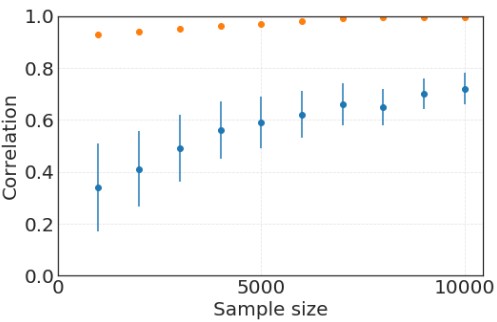
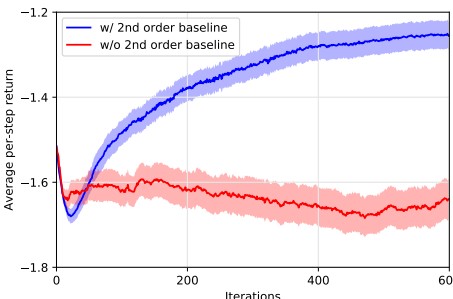

Figure 3: The correlation coefficients of the exact Hessian and the estimated Hessian generated from the multi-agent DiCE objective function with (orange) and without (blue) the second order baseline term $\mathcal{B}_{\Box}^{(2),i}$. The error bar shows the standard deviation.

Figure 4: The performance of the LOLA-DiCE algorithm on the IPD with (blue) and without (red) the new second order baseline. Shading indicates the error of the mean.

Figure 4 shows the performance of LOLA-DiCE with and without our second order baseline. We find that without the second order baseline agents fail to learn, resulting in an average per-step return of around $-1.6$, close to that of a random policy which achieves $-1.5$. Using a two times smaller batch size (32 vs 64), performance is comparable to what was achieved in the original work. However, our results using the first order baseline are much worse than what is reported in the original work, albeit at a smaller batch size. In communication with the authors we established that those results were produced by making the rewards at each timestep zero-mean within each batch, rather than relying on the first order baseline. In settings where the value function is mostly independent of the state, which happens to be the case in the IPD with a large $\gamma$, this simple trick can produce variance reduction similar to what we achieve with our second order baseline. However, this ad-hoc normalisation would fail in a setting with sparser rewards or, in general, in settings where the value function strongly depends on the current state.

## 5 CONCLUSION

Recent progress in multi-agent reinforcement learning and meta-learning has lead to a variety of approaches that employ second order gradient estimators. While these are easy to construct through the recently introduced DiCE objective (Foerster et al., 2018b), the high variance of second order gradient estimators has prevented their widespread application in practice. By reusing the DiCE formalism, we introduce a baseline for second order gradient estimators in stochastic computation graphs. Similar to DiCE, this baseline is automatically constructed from user-defined objectives using automatic differentiation frameworks, making it straightforward to use in practice. Our baseline does not change the expected value of any derivatives. We demonstrate empirically that our new baseline dramatically improves second order gradient estimation in a multi-agent task, reducing the required sample size by two orders of magnitude. We believe that low-variance second order gradient estimators will unlock a large variety of reinforcement learning and meta-learning applications in the future. Furthermore, we would like to extend the approach to deal with settings where the costs depend directly on the parameters. Lastly, we are interested in extending our framework to a baseline-generating term for any-order gradient estimators.

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

## A   APPENDIX

### A.1   FIRST ORDER GRADIENTS

**DiCE Objectice.**

$$\nabla_\theta \mathcal{L}_{\boxdot} = \sum_{c \in \mathcal{C}} \nabla_\theta \big( \boxdot(\mathcal{S}_c) \cdot c \big) = \sum_{c \in \mathcal{C}} c \cdot \nabla_\theta \boxdot(\mathcal{S}_c) + \sum_{c \in \mathcal{C}} \boxdot(\mathcal{S}_c) \cdot \nabla_\theta c$$

$$= \sum_{c \in \mathcal{C}} c \cdot \boxdot(\mathcal{S}_c) \sum_{w \in \mathcal{S}_c} \nabla_\theta \log p(w; \theta)) + \sum_{c \in \mathcal{C}} \boxdot(\mathcal{S}_c) \cdot \nabla_\theta c$$

$$\rightarrowtail \sum_{w \in \mathcal{S}} \sum_{c \in \mathcal{C}} \mathbf{1}_{(w \prec c)} c \cdot \nabla_\theta \log p(w; \theta) + \sum_{c \in \mathcal{C}} \nabla_\theta c$$

$$= \sum_{w \in \mathcal{S}} \left( \sum_{c \in \mathcal{C}_w} c \right) \cdot \nabla_\theta \log p(w; \theta) + \sum_{c \in \mathcal{C}} \nabla_\theta c$$

$$= \sum_{w \in \mathcal{S}} R_w \nabla_\theta \log p(w; \theta) + \sum_{c \in \mathcal{C}} \nabla_\theta c.$$

When the cost nodes does not depend on $\theta$ directly, we have $\nabla_\theta c = 0$. Thus,

$$\nabla_\theta \mathcal{L}_{\boxdot} \rightarrowtail \sum_{w \in \mathcal{S}} R_w \nabla_\theta \log p(w; \theta).$$

**Baseline Terms.**   For the first baseline term,

$$\nabla_\theta \mathcal{B}_{\boxdot}^{(1)} = - \sum_{w \in \mathcal{S}} b_w \nabla_\theta \boxdot(\{w\})$$

$$= - \sum_{w \in \mathcal{S}} b_w \boxdot(\{w\}) \nabla_\theta \log p(w; \theta)$$

$$\rightarrowtail - \sum_{w \in \mathcal{S}} b_w \nabla_\theta \log p(w; \theta). \tag{15}$$

We can consider a single term in (15),

$$\mathbb{E}[b_w \nabla_\theta \log p(w; \theta)] = b_w \sum_w p(w; \theta) \frac{\nabla_\theta p(w; \theta)}{p(w; \theta)}$$

$$= b_w \nabla_\theta \sum_w p(w; \theta)$$

$$= b_w \nabla_\theta 1 = 0.$$

According to the linearity of expectations, we have

$$\mathbb{E}[\nabla_\theta \mathcal{B}_{\boxdot}^{(1)}] \rightarrowtail 0.$$

For the second baseline term,

$$\nabla_\theta \mathcal{B}_{\boxdot}^{(2)} = \sum_{w \in \mathcal{S}'} b_w \Big[ - \big( 1 - \boxdot(\mathcal{S}_w) \big) \nabla_\theta \boxdot(\{w\}) - \big( 1 - \boxdot(\{w\}) \big) \nabla_\theta \boxdot(\mathcal{S}_w) \Big]$$

$$\rightarrowtail 0.$$

Obviously, $\mathbb{E}[\nabla_\theta \mathcal{B}_{\boxdot}^{(2)}] \rightarrowtail 0$.

### A.2   SECOND ORDER GRADIENTS

**DiCE Objective.**

$$\nabla_\theta^2 \mathcal{L}_{\boxdot} = \sum_{c \in \mathcal{C}} \nabla_\theta^2 \big( \boxdot(\mathcal{S}_c) \cdot c \big)$$

$$= \underbrace{\sum_{c \in \mathcal{C}} c \cdot \nabla_\theta^2 \boxdot(\mathcal{S}_c)}_{A} + \underbrace{\sum_{c \in \mathcal{C}} 2 \nabla_\theta c \cdot \nabla_\theta \boxdot(\mathcal{S}_c)}_{B} + \underbrace{\sum_{c \in \mathcal{C}} \boxdot(\mathcal{S}_c) \cdot \nabla_\theta^2 c}_{C}.$$

Next, we can evaluate terms $A$, $B$, and $C$,

$$A = \sum_{c \in \mathcal{C}} c \cdot \boxdot(\mathcal{S}_c) \Big[ \Big( \sum_{w \in \mathcal{S}_c} \nabla_\theta \log p(w; \theta) \Big)^2 + \sum_{w \in \mathcal{S}_c} \nabla_\theta^2 \log p(w; \theta) \Big]$$

$$\rightarrowtail \sum_{c \in \mathcal{C}} c \Big[ \sum_{w \in \mathcal{S}_c} (\nabla_\theta \log p(w\theta))^2 + 2 \sum_{w \in \mathcal{S}_c} \sum_{v \in \mathcal{S}_c, w \prec v} \nabla_\theta \log p(w; \theta) \cdot \nabla_\theta \log p(w; \theta) + \sum_{w \in \mathcal{S}_c} \nabla_\theta^2 \log_\theta p(w; \theta) \Big]$$

$$= \underbrace{\sum_{c \in \mathcal{C}} c \Big[ \sum_{w \in \mathcal{S}_c} \big( (\nabla_\theta \log p(w; \theta))^2 + \nabla_\theta^2 \log p(w; \theta) \big) \Big]}_{A_1}$$

$$+ 2 \underbrace{\sum_{c \in \mathcal{C}} c \Big[ \sum_{w \in \mathcal{S}_c} \sum_{v \in \mathcal{S}_c, w \prec v} \nabla_\theta \log p(w; \theta) \cdot \nabla_\theta \log p(v; \theta) \Big]}_{A_2},$$

$$A_1 = \sum_{w \in \mathcal{S}} \sum_{c \in \mathcal{C}} \mathbf{1}_{(w \prec c)} c \big[ (\nabla_\theta \log p(w; \theta))^2 + \nabla_\theta^2 \log p(w; \theta) \big]$$

$$= \sum_{w \in \mathcal{S}} \Big( \sum_{c \in \mathcal{C}_w} c \Big) \cdot \big[ (\nabla_\theta \log p(w; \theta))^2 + \nabla_\theta^2 \log p(w; \theta) \big]$$

$$= \sum_{w \in \mathcal{S}} R_w \frac{\nabla_\theta^2 p(w; \theta)}{p(w; \theta)},$$

$$A_2 = \sum_{c \in \mathcal{C}} \sum_{w \in \mathcal{S}} \sum_{v \in \mathcal{S}, w \prec v} \mathbf{1}_{v \prec c} \cdot \nabla_\theta \log p(w; \theta) \cdot \nabla_\theta \log p(v; \theta)$$

$$= \sum_{w \in \mathcal{S}} \sum_{v \in \mathcal{S}, w \prec v} \Big( \sum_{c \in \mathcal{C}_v} c \Big) \cdot \nabla_\theta \log p(w; \theta) \cdot \nabla_\theta \log p(v; \theta)$$

$$= \sum_{w \in \mathcal{S}} \nabla_\theta \log p(w; \theta) \Big[ \sum_{v \in \mathcal{S}, w \prec v} R_v \nabla_\theta \log p(v; \theta) \Big],$$

$$B = \sum_{c \in \mathcal{C}} 2 \nabla_\theta c \cdot \boxdot(\mathcal{S}_c) \Big[ \sum_{w \in \mathcal{S}_c} \nabla_\theta \log p(w; \theta) \Big]$$

$$\rightarrowtail \sum_{c \in \mathcal{C}} 2 \nabla_\theta c \Big[ \sum_{w \in \mathcal{S}_c} \nabla_\theta \log p(w; \theta) \Big],$$

$$C = \sum_{c \in \mathcal{C}} \boxdot(\mathcal{S}_c) \cdot \nabla_\theta^2 c \rightarrowtail \sum_{c \in \mathcal{C}} \nabla_\theta^2 c$$

As a result, we have

$$\nabla_\theta^2 \mathcal{L}_\boxdot \rightarrowtail \sum_{w \in \mathcal{S}} R_w \frac{\nabla_\theta^2 p(w; \theta)}{p(w; \theta)} + 2 \sum_{w \in \mathcal{S}} \nabla_\theta \log p(w; \theta) \Big[ \sum_{v \in \mathcal{S}, w \prec v} R_v \nabla_\theta \log p(v; \theta) \Big]$$

$$+ 2 \sum_{c \in \mathcal{C}} \nabla_\theta c \Big[ \sum_{w \in \mathcal{S}_c} \nabla_\theta \log p(w; \theta) \Big] + \sum_{c \in \mathcal{C}} \nabla_\theta^2 c.$$

When the cost nodes does not depend on $\theta$ directly, we have $\nabla_\theta c = 0$ and $\nabla_\theta^2 c = 0$. Thus,

$$\nabla_\theta^2 \mathcal{L}_\boxdot \rightarrowtail \sum_{w \in \mathcal{S}} R_w \frac{\nabla_\theta^2 p(w; \theta)}{p(w; \theta)} + 2 \sum_{w \in \mathcal{S}} \nabla_\theta \log p(w; \theta) \Big[ \sum_{v \in \mathcal{S}, w \prec v} R_v \nabla_\theta \log p(v; \theta) \Big].$$

**Baseline Terms.** For the first baseline term,

$$
\begin{aligned}
\nabla_\theta^2 \mathcal{B}_{\boxdot}^{(1)} &= -\sum_{w\in\mathcal{S}} b_w \nabla_\theta^2 \boxdot(\{w\}) \\
&= -\sum_{w\in\mathcal{S}} b_w \boxdot(a_t) \Big[ (\nabla_\theta \log p(w;\theta))^2 + \nabla_\theta^2 \log p(w;\theta) \Big] \\
&\rightarrowtail -\sum_{w\in\mathcal{S}} b_w \Big[ (\nabla_\theta \log p(w;\theta))^2 + \nabla_\theta^2 \log p(w;\theta) \Big] \\
&= -\sum_{w\in\mathcal{S}} b_w \Big[ \frac{(\nabla_\theta p(w;\theta))^2}{p(w;\theta)^2} + \frac{\nabla_\theta^2 p(w;\theta)}{p(w;\theta)} - \frac{(\nabla_\theta p(w;\theta))^2}{p(w;\theta)^2} \Big] \\
&= -\sum_{w\in\mathcal{S}} b_w \frac{\nabla_\theta^2 p(w;\theta)}{p(w;\theta)}.
\end{aligned}
\tag{16}
$$

We can consider a single term in (16),

$$
\begin{aligned}
\mathbb{E}\Big[ b_w \frac{\nabla_\theta^2 p(w;\theta)}{p(w;\theta)} \Big] &= b_w \sum_w p(w;\theta) \frac{\nabla_\theta^2 p(w;\theta)}{p(w;\theta)} \\
&= b_w \nabla_\theta^2 \sum_w p(w;\theta) \\
&= b_w \nabla_\theta^2 1 = 0.
\end{aligned}
$$

According to the linearity of expectations, we have

$$
\mathbb{E}[\nabla_\theta^2 \mathcal{B}_{\boxdot}^{(1)}] \rightarrowtail 0.
$$

For the second baseline term,

$$
\begin{aligned}
\nabla_\theta^2 \mathcal{B}_{\boxdot}^{(2)} &= \sum_{w\in\mathcal{S}'} b_w \Big[ -\nabla_\theta^2 \boxdot(\{w\}) \cdot \big(1 - \boxdot(\mathcal{S}_w)\big) - \big(1 - \boxdot(\{w\})\big) \cdot \nabla_\theta^2 \boxdot(\mathcal{S}_w) \\
&\qquad + 2\boxdot(\{w\}) \nabla_\theta \log p(w;\theta) \cdot \boxdot(\mathcal{S}_w) \sum_{v\in\mathcal{S}_w} \nabla_\theta \log p(v;\theta) \Big] \\
&\rightarrowtail 2\sum_{w\in\mathcal{S}'} b_w \nabla_\theta \log p(w;\theta) \Big[ \sum_{v\in\mathcal{S}_w} \nabla_\theta \log p(v;\theta) \Big] \\
&= 2\sum_{v\in\mathcal{S}} \sum_{w\in\mathcal{S}'} \mathbf{1}_{(v\prec w)} b_w \nabla_\theta \log p(w;\theta) \cdot \nabla_\theta \log p(v;\theta) \\
&= 2\sum_{v\in\mathcal{S}} \Big( \sum_{w\in\mathcal{S}_v, v\prec w} b_w \nabla_\theta \log p(w;\theta) \Big) \nabla_\theta \log p(v;\theta) \\
&= 2\sum_{v\in\mathcal{S}} \nabla_\theta \log p(v;\theta) \Big[ \sum_{w\in\mathcal{S}_v, v\prec w} \nabla_\theta \log p(w;\theta) \cdot b_w \Big].
\end{aligned}
\tag{17}
$$

Next, we consider the expectation of a single term in (17)

$$
\begin{aligned}
\mathbb{E}\Big[ \nabla_\theta \log p(v;\theta) \sum_{w\in\mathcal{S}_v, v\prec w} b_w \nabla_\theta \log p(w;\theta) \Big] &= \mathbb{E}\Big[ \mathbb{E}\Big[ \nabla_\theta \log p(v;\theta) \sum_{w\in\mathcal{S}_v, v\prec w} b_w \nabla_\theta \log p(w;\theta) \Big| v \Big] \Big] \\
&= \mathbb{E}\Big[ \nabla_\theta \log p(v;\theta) \sum_{w\in\mathcal{S}_v, v\prec w} b_w \mathbb{E}\big[ \nabla \log p(w;\theta) | v \big] \Big] \\
&= 0
\end{aligned}
$$

because $\mathbb{E}\big[ \nabla_\theta \log p(w;\theta) | v \big] = 0$ where $v \prec w$. According to the linearity of expectations, we can obtain that $\mathbb{E}[\nabla_\theta^2 \mathcal{B}_{\boxdot}] \rightarrowtail 0$.

### A.3 ARCHITECTURE AND HYPERPARAMETERS.

We use a tabular policy and value function, initialised from a normal distribution with unit variance and zero mean. The discount, $\gamma = 0.96$, and the episodes are truncated after 150 steps. Our experiments use a batch size of 32, a learning rate of 0.05 for the policy, and a look-ahead step size, $\alpha = 0.3$. To allow for proper variance reduction we train two value functions, one for the inner loop and one for the outer loop, which are pre-trained over 200 training steps. To ensure that the value function closely tracks the value under the changing policy we carry out 10 training steps of the value functions with a learning rate of 0.1 for each policy update.

