# OpenReview forum: "A Better Baseline for Second Order Gradient Estimation in Stochastic Computation Graphs"
_ICLR.cc/2019/Conference_

### Official Review · AnonReviewer1 · 2018-10-22
**Nicely written paper contributes useful trick; novelty too low for conference track, some correctness issues present**

**Rating:** 3
**Confidence:** 4

**Review:**

Overview:
This nicely written paper contributes a useful variance reduction baseline to make the recent formalism of the DiCE estimator more practical in application. I assess the novelty and scale of the current contribution as too low for publication at ICLR. Also, the paper includes a few incorrect assertions regarding the control variate framework as well as action-dependent baselines in reinforcement learning. Such issues reduce the value of the contribution in its current form and may contribute to ongoing misunderstandings of the control variate framework and action-dependent baselines in RL, to the detriment of variance reduction techniques in machine learning. I do not recommend publication at this time.

Pros:
The paper is well written modulo the issues discussed below. It strikes me as a valuable workshop contribution once the errors are addressed, but it lacks enough novelty for the main conference track.

Issues:

* (p.5) "R_w and b_w are positively correlated by design, as they should be for variance reduction of the first order gradients."

This statement is not true in general. Intuitively, a control variate reduces variance because when a single estimate of an expectation of a function diverges from its true value according to some delta, then, with high probability, some function strongly correlated with that function will also diverge with a similar delta. Such a delta might be positive or negative, so long as the error may be appropriately modeled as drawn from some symmetric distribution (i.e. is Gaussian).

Control variates are often estimated with an optimal scaling constant that depends on the covariance of the original function and its control variate. Due to the dependence on the covariance, the scaling constant flips sign as appropriate in order reduce variance for any delta. For more information, see the chapter on variance reduction and subsection on control variates in Sheldon Ross's textbook "Simulation."

The fact that a control variate appears to work despite this is not surprising. Biased and suboptimal unbiased gradient estimators have been shown to work well for reasons not fully explored in the literature yet. See, for example, Tucker et al.'s "Mirage of Action-Dependent Baselines", https://arxiv.org/abs/1802.10031.

Since the authors claim on page 6 that the baseline is positively correlated by design, this misunderstanding of the control variate framework appears to be baked into the baseline itself. I recommend the authors look into adaptively estimating an optimal scale for the baseline using a rolling estimator of the covariance and variance to fix this issue. See the Ross book cited above for full derivation of this optimal scale.

* The second error is a mischaracterization of the use and utility of action-dependent baselines for RL problems, on page 6: "We choose the baseline ... to be a function of state ... it must be independent of the action ...." and "it is essential to exclude the current action ... because the baselines ... must be independent of the action ... to remain unbiased." In the past year, a slew of papers have presented techniques for the use of action-dependent baselines, with mixed results (see the Mirage paper just cited), including two of the papers the authors cited.

Cons
* Much of paper revises the DiCE estimator results, arguing for and explaining again those results rather than referring to them as a citation.
* I assess the novelty of proposed contribution as too low for publication. The baseline is an extension of the same method used in the original paper, and does not generalize past the second order gradient, making the promising formalism of the DiCE estimator as infinitely differentiable still unrealizable in practice.
* The experiments are practically identical to the DiCE estimator paper, also reducing the novelty and contribution of the paper.

*EDIT:
I thank the authors for a careful point-by-point comparison of our disagreements on this paper so that we may continue the discussion. However, none of the points I identified were addressed, and so I maintain my original score and urge against publication. In their rebuttal, the authors have defended errors and misrepresentations in the original submission, and so I provide a detailed response to each of the numbered issues below:

(1) I acknowledge that it is common to set c=1 in experiments. This is not the same as the misstatements I cited, verbatim, in the paper that suggest this is required for variance reduction. My aim in identifying these mistakes is not to shame the authors (they appear to simply be typos) but simply to ensure that future work in this area begins with a correct understanding of the theory. I request again that the authors revise the cited lines that incorrectly state the reliance of a control variate on positive correlation. It is not enough to state that "everyone knows" what is meant when the actual claim is misleading.

(2) Without more empirical investigation, the authors' new claim that a strictly state-value-function baseline is a strength rather than a weakness cannot be evaluated. This may be the case, and I would welcome some set of experiments that establish this empirical claim by comparing against state-action-dependent baselines. The authors appear to believe that state-action-dependent baselines are never effective in reducing variance, and this is perhaps the central error in the paper that should be addressed. See response (3). Were the authors to fix this, they would necessarily compare against state-action-dependent baselines, which would be of great value for the community at large in settling this open issue.

(3) Action-dependent baselines have not been shown to be ineffective. I wish to strongly emphasize that this is not the conclusion of the Mirage paper, and the claim repeated in the authors' response (3) has not been validated empirically or analytically, and does not represent the state of variance reduction in reinforcement learning as of this note. I repeat a few key arguments from the Mirage paper in an attempt to dispel the authors' repeated misinterpretation of the paper.

The variance of the policy gradient estimator, subject to a baseline "phi," is decomposed using the Law of Total Variance in Eq (3) of the Mirage paper. This decomposition identifies a non-zero contribution from "phi(a,s)", the (adaptive or non-adaptive) baseline. The Mirage paper analyzes under what conditions such a contribution is expected to be non-negligible. Quoting from the paper:
"We expect this to be the case when single actions have a large effect on the overall discounted
return (e.g., in a Cliffworld domain, where a single action could cause the agent to fall of the cliff and suffer a large negative reward)."
Please see Sec. 3, "Policy Gradient Variance Decomposition" of the Mirage paper for further details.
The Mirage paper does indeed cast reasonable doubt on subsets of a few papers' experiments, and shows that the strong claim, mistakenly made by these papers, that state-action-dependence is always required for an adaptive control variate to reduce variance over state dependence, is not true.

It should be clear from the discussion of the paper to this point that this does _not_ imply the even stronger claim in "A Better Second Order Baseline" that action dependence is never effective and should no longer be considered as a means to reduce variance from a practitioner's point of view. Such a misinterpretation should not be legitimized through publication, as it will muddy the waters in future research. I again urge the authors to remove this mistake from the paper.

(4) I acknowledge the efforts of the authors to ensure that adequate background is provided for readers. This is a thorny issue, and it is difficult to balance in any work. Since this material represents a sizeable chunk of the paper and is nearly identical to existing published work, it leads me to lower the score for novelty of contribution simply by that fact. Perhaps the authors could have considered placing the extensive background materials in the appendix and instead summarizing them briefly in the body of the paper, leaving more room for discussion and experimental validation beyond the synthetic cases already studied in the DiCE paper.

(5), (6) In my review I provided specific, objective criteria by which I have assessed the novelty of this paper: the lack of original written material, and the nearly identical experiments to the DiCE paper. As I noted in response (4) above, this reduces space for further analysis and experimentation.

---

> ### Author Response · Authors · 2018-11-14
> **We strongly disagree with the main points raised, particularly the misconception around the optimal scaling constant ‘c’ and action-dependent baselines.**
>
> Thank you for your feedback. However, we strongly disagree with the main points of criticism, particularly the misconception around the optimal scaling constant ‘c’ and action-dependent baselines. Please see our detailed response to the individual points of the review below.
>
> 1) Re “positive vs negative correlation”: Of course covariates can be positively or negatively correlated. The fact that we say it should be positively correlated does not indicate a misunderstanding but merely reflects the fact that in our case c is set to 1 (see below why directly evaluating the optimal c is not practical in realistic settings).
>
> 2) Re “optimal scaling constant”: While it is well known that the optimal variance reduction depends on the covariance between the control variate and the estimator, this optimal factor is rarely used in practice for reinforcement learning due to the computational costs of doing one gradient estimate per entry in a batch. What is used in practice across the board for Deep RL (eg. A3C, PPO, IMPALA, etc) is the value-function based variance reduction, which we are enabling for higher order gradients through the DiCE formalism. The fact that our method only depends on the commonly used state-value-function for the baseline computation is a strength, not a weakness.
>
> 3) Re “independent of the action”: We will clarify this issue in the paper but the review misrepresents the facts on this point. Yes, there is a way to account for the bias introduced by an action-dependent baseline and in some cases this bias can be removed exactly. However, this is another method (like the optimal scaling factor mentioned above) that has not been shown to work in practice. In fact the very paper cited by the reviewer (the ‘mirage of action dependent baselines’), concludes that from a practitioner's point of view there currently is no reason to consider action dependent baselines. Our submission extends the utility of value functions to provide action-independent variance reduction for higher order gradients.
>
> 4) Re “revises DiCE formalism”: Thank you for this suggestion. Prior to submission, we carefully considered how our contribution and decided that in order for the paper to be self-contained, Stochastic Computation Graphs as well as the DiCE formalism should be explained clearly in the Background section. To highlight the delta to prior work, we clearly separated out the revision of Stochastic Computation Graphs and the DiCE formalism (Section 2) from the Method (Section 3).
>
> 5) Re “novelty too low.. does not generalize past the second order gradient”: This is obviously a subjective claim. However, note that second order gradients are a key use-case in meta-learning and multi-agent RL and as such the new baseline has the potential to unlock a large number of applications and is of key importance to the community (in fact we are aware of one other research group that has already started experimentation with this baseline). Also, just as the 1st order variance reduction term contributes to a lower 2nd order variance, our new 2nd order baseline also acts as a variance reduction for higher order gradient estimators, although we did not quantify the impact experimentally (since 2nd order is the most relevant for current research).
>
> 6) Re “experiments are identical”: This is also a strength, not a weakness: for the sake of reproducibility between the original DiCE and the new baseline, it is crucial to use the same setting. Also, note that his paper is about proposing a new tool, rather than demonstrating full solutions to novel applications. Experimental results are provided as proof-of-principle and are not the main point of the paper. Clearly, the experimental results support the utility of the new baseline compared to a previously published result.

---

> ### Author Response · Authors · 2018-11-28
> **A detailed review that partially misses the point due to a miss understanding of the purpose of the paper.**
>
> First of all, we would like to thank you for taking the time to review the paper and for taking the time to reply to our comments. This is appreciated, especially at a busy time of the year like this.
>
> @1) This is fair: While from a practitioner’s perspective ‘c = 1’, from an educational point we agree that mentioning the optimal ‘c’ is valuable and we’ll include this in the next revision of the paper. We had omitted this since it seemed irrelevant from a practical point of view, but you are right that it is useful background.
>
> @2)-5): All of these points indicate a miss-understanding of the paper’s main point, which we will emphasise more in a revision:
> By “better baseline” we mean literally “better than current DiCE baseline”. So the point is not a comparison between action-dependent and state-dependent baselines, but simply making a ‘better baseline’ easily available.
> We agree that investigating action-dependent baselines is a fascinating research area. However, that’s also the main reason why we do not focus on them in our work: The point of this paper is to make methods that have been proven to work (and are commonly being used), more easily available to practitioners.
> This fits in nicely with the narrative of DiCE overall: The main point of DiCE is to facilitate the development and deployment of methods that require higher order gradients. Note that higher order gradients here are not the subject of the research, but merely a required tool. This process should be pursued in parallel to the development, investigation, and analysis of different higher order estimators and variance reduction techniques.
> We also strongly disagree with the statement that the baseline should be tested on novel settings to increase the novelty of the paper: Reproducibility is absolutely vital for scientific progress, especially for the development of tools (such as DiCE and its baselines).
>
> So I think what it comes down to is this: Are higher order gradient estimators well enough developed so that they can be used for practical applications as a tool rather than having to be the subject of research itself? Our belief is yes and this submission constitutes an important step in that direction. Reducing the sample-requirements by a factor of 100x should not be just brushed aside, even if it's on a 'toy' problem.
>
> For future work we do agree that it would be great to extend our formalism for the baseline to include the option of having action-dependent baselines and others.

---

### Official Review · AnonReviewer3 · 2018-11-05
**An important direction motivated by recent need for second-order gradient estimation, but need to verify its advantages more thoroughly**

**Rating:** 6
**Confidence:** 3

**Review:**

In this paper, the author proposed a better control variate formula for second-order Monte Carlo gradient estimators, based on a special version of DiCE (Foerster et al, 2018).  The motivation and the main method is easy to follow and the paper is well written.  The author followed the same experiments setting as DiCE, numerically verifying the advantages of the newly proposed baseline, which can estimate the Hession accurately.

The work is essentially important due to the need for second-order gradient estimation for meta-learning (Finn et al., 2017) and multi-agent reinforcement learnings.  However, the advantage of the proposed method is not verified thoroughly. The only real application demonstrated in the paper, can be achieved the same performance as the second-order baseline using a simple trick.  Since this work only focuses on second-order gradient estimations, I think it would be better to verify its advantages in various scenarios such as meta-learning or sparse reward RL  as the author suggested in the paper.

Finn, Chelsea, Pieter Abbeel, and Sergey Levine. "Model-agnostic meta-learning for fast adaptation of deep networks." ICML 2017.
Foerster, Jakob, et al. "DiCE: The Infinitely Differentiable Monte-Carlo Estimator." ICML 2018.

---

> ### Author Response · Authors · 2018-11-14
> **Thank you for the positive review, we’re excited about applying this tool to larger problems in future work.**
>
> Many thanks for the review. While we agree that more experimental validation would have value, this paper is primarily proposing a novel method, which is validated both through proof-of-principle experiments, but, also, and more importantly, theoretically. Furthermore, since we uploaded the paper to OpenReview, another research group has already started experimenting with the new baseline.

---

### Official Review · AnonReviewer5 · 2018-11-14
**A paper on an important topic, but the contribution is not very significant**

**Rating:** 5
**Confidence:** 4

**Review:**

Thank you for an interesting read.

This paper extends the recently published DiCE estimator for gradients of SCGs and proposed a control variate method for the second order gradient. The paper is well written. Experiments are a bit too toy, but the authors did show significant improvements over DiCE with no control variate.

Given that control variates are widely used in deep RL and Monte Carlo VI, the paper can be interesting to many people. I haven't read the DiCE paper, but my impression is that DiCE found a way to conveniently implement the REINFORCE rules applied infinite times. So if I were to derive a baseline control variate for the second or higher order derivatives, I would "reverse engineer" from the exact derivatives and figure out the corresponding DiCE formula. Therefore I would say the proposed idea is new, although fairly straightforward for people who knows REINFORCE and baseline methods.

For me, the biggest issue of the paper is the lack of explanation on the choice of the baseline. Why using the same baseline b_w for both control variates? Is this choice optimal for the second order control variate, even when b_w is selected to be optimal for the first order control variate? The paper has no explanation on this issue, and if the answer is no, then it's important to find out an (approximately) optimal baseline for this second order control variate.

Also the evaluation seems quite toy. As the design choice of b_w is not rigorously explained, I am not sure the better performance of the variance-reduced derivatives generalises to more complicated tasks such as MAML for few-shot learning.

Minor:
1. In DiCE, given a set of stochastic nodes W, why did you use marginal distributions p(w, \theta) for a node w in W, instead of the joint distribution p(W, \theta)? I agree that there's no need to use p(S, \theta) that includes all stochastic nodes, but I can't see why using marginal distribution is valid when nodes in W are not independent.

2. For the choice of b_w discussed below eq (4), you probably need to cite [1][2].

3. In your experiments, what does "correlation coefficient" mean? Normalised dot product?

[1] Mnih and Rezende (2016). Variational inference for Monte Carlo objectives. ICML 2016.
[2] Titsias and Lázaro-Gredilla (2015). Local Expectation Gradients for Black Box Variational Inference. NIPS 2015.

---

> ### Author Response · Authors · 2018-11-28
> **Thank you for the relevant references and the review. We disagree on relevance (as you can imagine).**
>
> “I would "reverse engineer" from the exact derivatives and figure out the corresponding DiCE formula.”
> There are two separate but related challenges: First of all, you need to formulate the correct baseline for the 2nd order derivatives. In particular we wanted to make sure that this baseline can be constructed using the standard state-value function.
> Secondly, this baseline needs to be constructed via a combination of DiCE operators, such that it can be included in the original objective. In other words, it needs to leave the evaluation of both, the original objective and the first order gradient, unchanged, but then also generate the correct terms for the 2nd order variance reduction when differentiated twice. This is non-trivial.
>
> “b_w,.. Is this choice optimal for the second order control variate”:
> The main application of the DiCE formalism is within the context of Reinforcement Learning. In Reinforcement Learning, b_w is simply the state value-function, V(s). While this is not an optimal baseline, it is the best *practical* baseline based upon applications. It is used by state-of-the-art algorithms such as PPO (https://arxiv.org/abs/1707.06347), A3C (https://arxiv.org/pdf/1602.01783.pdf) and others.
> Our 2nd order baseline is the extension of this ‘good enough’ baseline to higher order terms. As such it is not an optimal baseline, but a ‘good enough’ and easy to implement one.
>
> “design choice of b_w is not rigorously explained,”:
> Our focus is on having a variance reduction baseline which keeps the estimator unbiased. As such the terms need to be of the form b_w as described in the paper. We’ll clarify this further as appropriate. We respectfully disagree that MAML is a more complex task. LOLA has the same properties in terms of differentiating through the learning step of an agent. One major difference here is the continuous vs discrete action space.
>
> “why using marginal distribution is valid when nodes in W are not independent.”:
> The nodes w are indeed independent when conditioned on their causes. Intuitively, you can think of this as the actions being sampled iid once you condition on the states.
>
> “Cite [1], [2]”:
> Many thanks, we’ll update the paper to include these references!
>
> "correlation coefficient":
> -Yes.

---

### Official Review · AnonReviewer4 · 2018-11-15
**Interesting paper, could push it further**

**Rating:** 6
**Confidence:** 3

**Review:**

This paper extends the "infinitely differentiable Monte Carlo gradient estimator" (or DiCE) with a better control variate baseline for reducing the variance of the second order gradient estimates.

The paper is fairly clear and well written, and shows significant improvements on the tasks used in the DiCE paper.

I think the paper would be a much stronger submission with the following improvements:

- More explanation/intuition for how the authors came up with their new baseline (eq. (8)). As the paper currently reads, it feels as if it comes out of nowhere.
- Some analysis of the variance of the two terms in the second derivative in eq. (11). In particular, it would be nice to show the variance of the two terms separately (for both DiCE and this paper), to show that the reduction in variance is isolated to the second term (I get that this must be the case, given the math, but would be nice to see some verification of this). Also I do not have good intuition for which of these two terms dominates the variance.
- I appreciate that the authors tested their estimator on the same tasks as in the DiCE paper, which makes it easy to compare them. However, I think the paper would have much more impact if the authors could demonstrate that their estimator allows them to solve new, more difficult problems. Some of these potential applications are discussed in the introduction, it would be nice if the authors could demonstrate improvements in those domains.

As is, the paper is still a nice contribution.

---

> ### Author Response · Authors · 2018-11-28
> **Thank you for an encouraging and insightful review**
>
> Thank you for an encouraging and insightful review. We address specific points below.
>
> “More explanation/intuition”:
> We will add further intuition regarding the construction of the 2nd order baseline. The basic derivation is currently provided in the appendix on the bottom of page 11: When the DiCE objective is differentiated twice, the resulting terms can be rewritten as a double summation over nodes in the graph, with the inner sum containing an R_v (ie. sum of downstream costs). Importantly, the first order baseline does not provide a variance-reduction term for the R_v in term A^2. To do so we need a term that is the same as the A^2 term but contains -b_w instead of R_v, after double differentiation. That’s how we constructed the 2nd order baseline. We fully agree that this part should be explained more clearly in the paper and will do so in the next version.
>
> “show that the reduction in variance is isolated to the second term”:
> In fact, we already show this in the paper. In Figure 3, we compare the performance of the original DiCE objective (including the first order baseline) with the DiCE objective including the 1st and 2nd order baselines. We will make this point more clear in the text.
>
> “ solve new, more difficult problems”:
> This is a great idea which we hope to address in future work. We also believe that making this method broadly available will encourage other research groups to use the tool to solve new problems.

---

### Meta-Review · Area_Chair1 · 2018-12-14
**Good work but some critical issues need to be addressed**

**Confidence:** 4
**Recommendation:** Reject

**Metareview:**

This paper extends the DiCE estimator with a better control variate baseline for variance reduction.
The reviewers all think the paper is fairly clear and well written. However, as the reviews and discussion indicates,  there are several critical issues, including lack of explanation of the choice of baseline, the lack more realistic experiments and a few misleading assertions.  We encourage the authors to rewrite the paper to address these criticism. We believe this work will make a successful submission with proper modification in the future.